# Discovering and Overcoming Limitations of Noise-engineered Data-free Knowledge Distillation

**Piyush Raikwar**[*]
ABV-IIITM, Gwalior, India
imt_2017062@iiitm.ac.in

**Deepak Mishra**
IIT Jodhpur, India
dmishra@iitj.ac.in

## Abstract

Distillation in neural networks using only the samples randomly drawn from a Gaussian distribution is possibly the most straightforward solution one can think of for the complex problem of knowledge transfer from one network (teacher) to the other (student). If successfully done, it can eliminate the requirement of teacher's training data for knowledge distillation and avoid often arising privacy concerns in sensitive applications such as healthcare. There have been some recent attempts at Gaussian noise-based data-free knowledge distillation, however, none of them offer a consistent or reliable solution. We identify the shift in the distribution of hidden layer activation as the key limiting factor, which occurs when Gaussian noise is fed to the teacher network instead of the accustomed training data. We propose a simple solution to mitigate this shift and show that for vision tasks, such as classification, it is possible to achieve a performance close to the teacher by just using the samples randomly drawn from a Gaussian distribution. We validate our approach on CIFAR10, CIFAR100, SVHN, and Food101 datasets. We further show that in situations of sparsely available original data for distillation, the proposed Gaussian noise-based knowledge distillation method can outperform the distillation using the available data with a large margin. Our work lays the foundation for further research in the direction of noise-engineered knowledge distillation using random samples.

## 1 Introduction

Deep neural networks are an excellent choice for various real-world computer vision tasks. However, their high computational and space complexity becomes a bottleneck when it comes to deployment on resource-constrained devices. To address this issue, the idea of Knowledge Distillation (KD), proposed by [1], has recently gained significant attention. It enables the transfer of knowledge from a large neural network to a comparatively smaller one. The lightweight small neural network, often referred to as student, gains the information learned by the over-parameterized large model, referred to as teacher, and delivers a similar or sometimes even better performance [2].

Traditionally, in KD, original data is passed through the accurate teacher model to collect soft labels, and subsequently, these soft labels are used to supervise the student [3]. However, in certain applications such as medical, it often becomes impossible to release the original data due to security concerns, sensitive information, or storage restrictions. In contrast, a trained model (teacher) usually does not hold a specific record or specific piece of information thus, no certainty or validity of the information extracted through a neural network can be guaranteed, thereby making them safer for public release. However, these models certainly contain information in the parameterized form for performing the downstream tasks, such as classification or object detection.

---

[*]Work done during an internship at IIT Jodhpur. Presently at CERN, Geneva, CH (piyush.raikwar@cern.ch)

36th Conference on Neural Information Processing Systems (NeurIPS 2022).

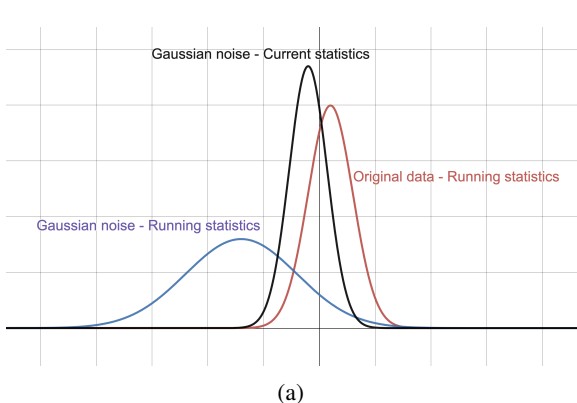
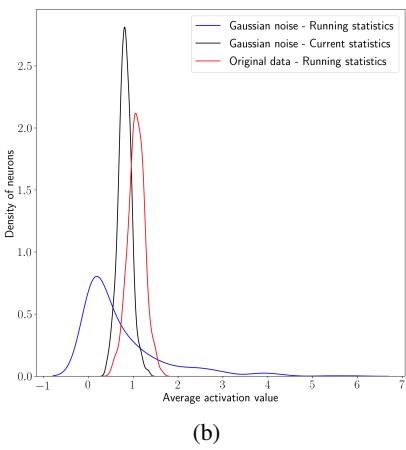

| (a) | (b) |

Figure 1: (a) Illustration of the effect of BatchNorm state in a teacher network. Each curve represents the distribution of average activations of neurons belonging to some intermediate layer in a pretrained teacher network. These are obtained when a specific type of input (original data or Gaussian noise) is fed to the teacher network under different states of BatchNorm layers. The red curve is the oracle representing activation distribution when original data is fed to the teacher. The other two curves are obtained when Gaussian noise is fed as the input, and when BatchNorm layers use the statistics of the original data (blue), and when it uses the current mini-batch statistics (black). (b) Distribution of average activation values of all neurons in 'avgpool' layer of ResNet-34 teacher observed in case of CIFAR10 experiments.

Due to these reasons, recently there has been an interest in data-free distillation - that is, distilling knowledge from one neural network to the other without using any data. Majority of methods in this direction aim to create synthetic images either by optimizing the randomly generated samples [4, 5, 6, 7, 8] or by training a generative network [9, 10], which often is computationally expensive. This motivates the research community to ask the question *Can distillation be done by using the samples randomly drawn from a known distribution such as standard normal distribution?* To best of our knowledge, we for the first time provide a satisfactory answer to the question and show that yes, it is possible to do data-free KD by only using images that are randomly sampled from a Gaussian distribution. Our experiments show that a considerably better performance is possible using only the Gaussian noise as the input, in contrast to what prior works ([9], [10]) have shown for this particular setting. We identify that the key limitation, which the prior works failed to identify is the covariate shift in the distribution of hidden layer activations of the teacher network caused by feeding Gaussian noise instead of the expected original data. As shown in Figure 1a, the activation distribution when input is Gaussian noise (blue) is shifted, thus no longer aligns with the activation distribution when input is original data (red). We propose a simple yet surprisingly effective solution to reduce this shift by using the current statistics instead of using statistics of the original data (running statistics) in the teacher's batch normalization (BatchNorm) layers and thus distill the knowledge from teacher into student network using just the Gaussian noise. Our main contributions are as follows:

- We explain how covariate shift in the teacher network interferes with Gaussian noise-based KD and propose a simple solution to eliminate it, thus enable knowledge transfer using Gaussian noise.

- We show how this phenomena of covariate shift occurs in the student network as well and mitigate it in a similar way.

- We obtain student performance close to the teacher without using any original or synthetic data. We further show that in the presence of limited data, the proposed approach outperforms the traditional distillation method with a large margin.

- Our work counters the commonly accepted notion of requiring realistic training data for data-free KD. Thus, we lay the foundation for noise-based data-free KD.

## 2 Related work

**Knowledge Distillation:** KD is the transfer of knowledge from one neural network to the other. First successful implementation of knowledge transfer was shown by [11] for the purpose of model compression. Later, the term knowledge distillation was popularized by [1], where the authors explored so called "dark knowledge" of a teacher model which is transferred to a student model with the help of soft labels. Since then, the domain of KD has witnessed many interested ideas related to improved distillation strategies such as teacher assistant [12] and self-distillation [13]. KD approaches for performing various downstream tasks such as object detection [14], semantic segmentation [15], and also the applications in different modalities, including speech [16] and graph [17] have been explored. A detailed survey on KD is recently reported by [2].

**Data-free Knowledge Distillation:** The problem of data-free KD, i.e., the constraint that the original dataset, which was used to train and supervise the teacher, is unavailable for distillation makes the tasks naturally challenging. This often is encountered in privacy-sensitive settings, where the original dataset cannot be shared. There are various prior works with a similar context in mind. Most of these methods first construct a set of synthetic images that resembles original data, then distill the knowledge of the teacher network into the student network using that synthetic data [4, 5, 6, 8, 9, 10]. Some approaches synthesize images one-by-one or batch-by-batch, whereas some train a generator model to gain access to limitless synthetic data. For example, [4] stored activation records of the teacher model when original data is feed-forwarded through it, and then later used it to create synthetic images by optimizing random noise using gradient descent. [5] introduced DeepInversion intending to create realistic images, which makes use of feature map and BatchNorm statistics to optimize random images into high-fidelity images. [18] takes a different approach to simultaneously generate images, as well as train the student. They introduced Zero-shot knowledge transfer, which utilized adversarial learning between the generator model and the student model. Their intuition was to search for images on which the student poorly matches the teacher and then train the student with those.

**Gaussian Noise-based Knowledge Distillation:** KD using Gaussian noise has been presented as a baseline numerous times in prior works, which resulted in a student accuracy slightly better than random chance. Although a decent accuracy was reported on MNIST (88% [9], 75% [19]), probably due to smaller networks and less complex dataset, on more challenging datasets like CIFAR10, a poor performance (14% [5], 15% [9], 11% [10], 10% [19]) was observed. Apart from these, [6] presented an interesting, although a highly constrained scenario where the student had an identical architecture to the teacher and student's weights were initialized as quantized values of teacher's weights. Further when the BatchNorm layer parameters from teacher are copied to student and kept frozen, Gaussian noise based distillation resulted in good performance. Moreover, instead of training the student network end-to-end as in traditional KD, [6] partitioned the student and teacher networks across BatchNorm layers resulting in multiple teacher-student pairs of blocks and distilled each pair of blocks separately. Although this approach works, it cannot be used for a pair of architectures not having similar block structures and is unnecessarily complex.

In contrast to all above, our aim is to design a simple approach for Gaussian noise-based data free distillation. This would not only relax the constraints of approaches such as [6] but also save the efforts required in creating synthetic dataset.

## 3 Methodology

### 3.1 Preliminaries

**Knowledge Distillation:** KD is achieved by training the student network using the output of the teacher network as soft labels instead of using the ground truth labels [3]. This results in matching of the output probability distributions of both the networks for a given input data. Formally, for a k-class classification, given a dataset $D$ that consists of data samples $x$, a teacher network $T$ trained on $D$, and a student network $S$ with randomly initialized weights, the following cost function is optimized:

$$L_{KD} = L_{\mathcal{XE}}(S(x), T(x)), \tag{1}$$

where $L_{\mathcal{XE}}(.,.)$ is the standard cross-entropy term.

**Batch Normalization:** BatchNorm was introduced by [20] that made the training of deep neural networks more stable and allowed to use larger learning rates, hence faster convergence. Further, it makes neural networks more robust to initialization schemes and changes in the learning rate. BatchNorm first normalizes the output of the hidden layer (input to BatchNorm layer) using the first (mean) and the second (variance) statistical moments of the current batch. Then, these normalized outputs are shifted and scaled according to the two trainable parameters $\gamma$ and $\beta$, respectively. These parameters are tuned as any other neural network parameters, and they help in choosing the optimum distribution as the input for the next layer, thus regaining expressiveness lost in the normalization step.

Formally, let $h$ be the activations of a neuron in the previous layer (input to the BatchNorm layer), $m$ be the batch-size, and $\epsilon$ be a small constant used for numerical stability. Then,

$$\overline{h_i} = \frac{h_i - \mu_B}{\sqrt{\sigma_B^2 + \epsilon}} \tag{2}$$

$$\widehat{h_i} = \gamma \times \overline{h_i} + \beta \tag{3}$$

where $\mu_B = \frac{1}{m}\sum_i h_i$ and $\sigma_B^2 = \frac{1}{m}\sum_i(h_i - \mu_B)^2$. The $\overline{h_i}$ and $\widehat{h_i}$ show normalized and subsequently shifted values of $h_i$. Above mentioned activation normalization using mini-batch or current statistics results in improved training. During evaluation or inference, the batch statistics $\mu_B$ and $\sigma_B$ are replaced with running or population statistics, $\mu_r = E_{\mathcal{B}}[\mu_B]$ and $\sigma_r = \frac{m}{m-1}E_{\mathcal{B}}[\sigma_B^2]$, where $E_{\mathcal{B}}[.]$ denotes expectation over training mini-batches. Accordingly, the normalization steps are modified as follows:

$$\overline{h_i} = \frac{h_i - \mu_r}{\sqrt{\sigma_r^2 + \epsilon}} \tag{4}$$

where $\mu_r = \alpha\mu_r + (1-\alpha)\mu_B$ and $\sigma_r = \alpha\sigma_r + (1-\alpha)\sigma_B$. Here $\alpha$ is a smoothing factor, which is there to prioritize statistics of the current batch.

## 3.2 Distillation using Gaussian Noise

We consider the most straightforward approach for KD in the absence of original data, just using random images sampled from a Gaussian distribution as an input to teacher and student network. This precondition to use Gaussian noise for KD might seem to make the task of distillation almost impossible since the input Gaussian distribution is notably different from the original data distribution, and does not contain any structural properties or representation as the original data. We hypothesize that, for successful KD, across two input data: first using which the teacher is trained, and second using which student is trained, having similar output distributions throughout the teacher network's hidden layers is more important than having the same structural or representation properties at the input; thus, leading to the extraction of meaningful information from the teacher network. To understand this better let us consider a trivial case where samples drawn from a standard Gaussian distribution ($\mathcal{N}(0,1)$) are passed through a BatchNorm layer optimized for some real data $D$. As during distillation only student network is trained, BatchNorm performs normalization of input according to (4). Consider the following statistics of the normalized signal.

$$E_{h_i \sim \mathcal{N}(0,1)}[\overline{h_i}] = \frac{E_{h_i \sim \mathcal{N}(0,1)}[h_i] - \mu_r}{\sqrt{\sigma_r^2 + \epsilon}} = \frac{0 - \mu_r}{\sqrt{\sigma_r^2 + \epsilon}} \tag{5}$$

$$\text{Var}[\overline{h_i}] = \frac{\text{Var}[h_i]}{\sigma_r^2 + \epsilon} = \frac{1}{\sigma_r^2 + \epsilon} \tag{6}$$

Equation (5) and (6) clearly show the deviation or covariate shift in the output of BatchNorm when Gaussian noise is passed instead of the original data. This suggest that output of BatchNorm should be shifted by $\frac{\mu_r}{\sqrt{\sigma_r^2 + \epsilon}}$ and scaled by $\sigma_r^2 + \epsilon$ before feeding it to the subsequent layers. However, this is not trivial for intermediate hidden layers of the teacher network as the activation statistics are unknown. Alternatively, as explained below, we may allow BatchNorm to simply use current statistics instead of running statistics as the corrections required are nothing but the factors of running statistics.

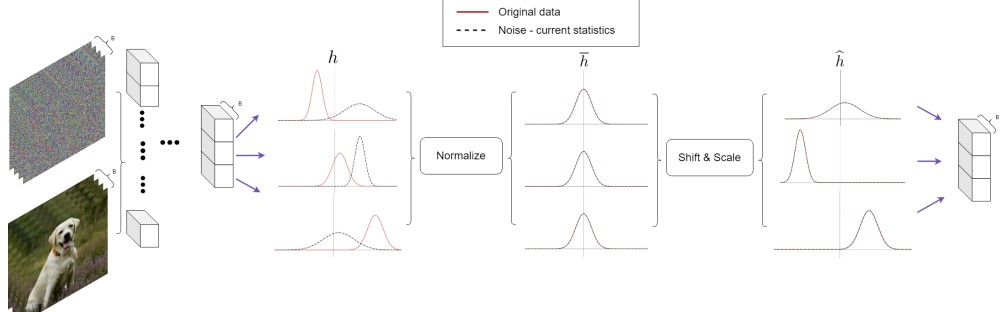

Figure 2: Inside the BatchNorm layer. The figure depicts an expanded view of a BatchNorm layer when each of original data and Gaussian noise is fed to it. Initially, the activation distribution of the hidden layer (input to BatchNorm) is very different in the case of Gaussian noise (black) compared to original data (red). We use the current statistics derived from Gaussian noise instead of data derived running statistics in the normalization step to mitigate the covariate shift. The rest of the shifting and scaling follows as usual[2]. Our method ensures that keeping $\gamma$ and $\beta$ constant will yield similar $P(\widehat{h})$ irrespective of the input distribution $P(h)$.

### 3.2.1 Mitigating the Shift

The solution to mitigate the observed shift and successfully perform distillation using Gaussian noise is to effectively make use of the BatchNorm layers. Our goal is to align the activation distribution of hidden layers throughout the teacher network when Gaussian noise is used as an input to that when original data is used as an input. We consider the case of a BatchNorm layer placed after a hidden layer $\mathcal{H}$ within a teacher network. Let $B$ represent a batch of images sampled from original dataset $D$, $G$ be a batch of random samples sampled from a Gaussian distribution, $\boldsymbol{h}$ be the vector of activations for some neuron $n$ belonging to the hidden layer $\mathcal{H}$ when inputs are fed to the teacher with a batch size $m$, and $P(\boldsymbol{h})$ be the activation distribution of that $n^{th}$ neuron over the batch. Due to the covariate shift, we observe that: $P(\boldsymbol{h}|G) \neq P(\boldsymbol{h}|B)$. Generally, during KD, the BatchNorm layers in the teacher network are setup for inference, hence use running statistics ($\boldsymbol{\mu}_r$, $\boldsymbol{\sigma}_r$), i.e., Equation (4) to calculate the normalized activations $\overline{\boldsymbol{h}}$. This leads to:

$$P(\overline{\boldsymbol{h}}|G, \boldsymbol{\mu}_r, \boldsymbol{\sigma}_r) \neq P(\overline{\boldsymbol{h}}|B, \boldsymbol{\mu}_r, \boldsymbol{\sigma}_r)$$
$$\text{Thus, } P(\widehat{\boldsymbol{h}}|G, \boldsymbol{\mu}_r, \boldsymbol{\sigma}_r, \boldsymbol{\gamma}, \boldsymbol{\beta}) \neq P(\widehat{\boldsymbol{h}}|B, \boldsymbol{\mu}_r, \boldsymbol{\sigma}_r, \boldsymbol{\gamma}, \boldsymbol{\beta})$$

Here, $\widehat{\boldsymbol{h}}$ is the scaled and shifted activation vector, which is the output of the BatchNorm layer for the $n^{th}$ neuron, as defined in Equation (4). This drifted $\widehat{\boldsymbol{h}}$ propagates to the next hidden layer and eventually to the output layer without any meaningful information being passed from the hidden layer $\mathcal{H}$. Therefore, for KD using Gaussian noise, we propose to use current batch statistics ($\boldsymbol{\mu}_B$, $\boldsymbol{\sigma}_B$) for the normalization step, i.e., Equation (2) in all BatchNorm layers of the teacher network. The intuition is to align the two drifted distributions at the normalization step. This makes sense only because $\boldsymbol{\gamma}$ and $\boldsymbol{\beta}$ have the inherent information about the distribution that the next layer expects and are designed to shift a standard normal distribution. Therefore, even though $P(\boldsymbol{h}|G) \neq P(\boldsymbol{h}|B)$ holds true:

$$P(\overline{\boldsymbol{h}}|G, \boldsymbol{\mu}_B, \boldsymbol{\sigma}_B) = P(\overline{\boldsymbol{h}}|B, \boldsymbol{\mu}_B, \boldsymbol{\sigma}_B)$$
$$\approx \mathcal{N}(0, 1)$$
$$\text{Hence, } P(\widehat{\boldsymbol{h}}|G, \boldsymbol{\mu}_B, \boldsymbol{\sigma}_B, \boldsymbol{\gamma}, \boldsymbol{\beta}) = P(\widehat{\boldsymbol{h}}|B, \boldsymbol{\mu}_B, \boldsymbol{\sigma}_B, \boldsymbol{\gamma}, \boldsymbol{\beta})$$

This reduces the covariate shift due to a different input distribution from the expected one at every stage (BatchNorm layer) in the teacher network. It obtains $P(\widehat{\boldsymbol{h}})$ similar to that of using original data irrespective of $P(\boldsymbol{h})$, and hence leads to the transfer of meaningful information. Our approach is visually explained in Figure 2, where we show an expanded view of a BatchNorm layer for three neurons. It shows how our proposed method effectively uses the normalization step to align the two activation distributions: (i) when Gaussian noise is fed as inputs (ii) when original data is fed as

---

[2]To understand this further, consider a toy example presented in the Appendix A.1.

inputs. Note that the covariate shift discussed throughout this paper is not the same as the internal covariate shift hypothesized by [20], even though ours might be happening in the hidden layers of the neural network as well. The internal covariate shift occurs due to constantly changing network parameters in the preceding layer, thus affecting the input distribution for the following layer. On the other hand, we consider the shift due to change in the input that is being fed to the network, which in turn affects the input distribution of hidden layers as well.

### 3.3 Inference

Similar to how a teacher network trained using original data distribution cannot directly handle the Gaussian noise, the student network trained using Gaussian noise will also not be able to provide correct output during test time, as the running statistics of BatchNorm layers in the student expect Gaussian noise activation patterns. The inference is, therefore, done using current statistics. Moreover, each mini-batch needs to be independent and identically distributed, and large enough to approximate the test data distribution. Usually, it should not matter how the input is arranged during the inference, therefore, a simple workaround to this problem is to first adjust the running statistics of the student network catering to the test data by feed-forwarding some amount of shuffled test data before evaluating the student network. Note that no back-propagation or optimization steps are required as only the running statistics need to be adjusted. We summarize the training and evaluation phases of our approach in Algorithm 1 and Algorithm 2 respectively.

---

**Algorithm 1** Training - KD

**Requires:** pretrained teacher $T(.)$
**Initialize:** student $S(.;\theta)$ with parameters $\theta$
**for** $B$ **in** $1, 2, ..., \mathcal{B}_1$ **do**
  $G \sim \mathcal{N}(0, 1)$
  $y_T \leftarrow T(G|\mu_B, \sigma_B)$
  $y_S \leftarrow S(G|\theta, \mu_B, \sigma_B)$
  $\theta \leftarrow \theta - \eta \frac{\partial L_{KD}}{\partial \theta}$
**end for**

---

**Algorithm 2** Evaluation

**Requires:** pretrained student $S(.;\theta)$
**for** $B$ **in** $1, 2, ..., \mathcal{B}_2$ **do**
  $X \sim D$
  $y_S \leftarrow S(X|\theta, \mu_B, \sigma_B)$
  $y_{label} \leftarrow argmax(y_S)$
**end for**

---

## 4 Experiments

In this section, we discuss the experiments we performed on various datasets in combination with different standard neural network architectures to demonstrate the working of our proposed method. We use CIFAR10 [21] as a proof of concept to establish the validity of our method, and for certain ablations. We further test our proposed approach on SVHN [22], CIFAR100, and Food101 [23]. Note that, in all the experiments, for Gaussian noise, we generate the image pixels from Gaussian distribution having mean, $\mu = 0$ and variance, $\sigma^2 = 1$.

### 4.1 State of BatchNorm

In this experiment, we discuss the effect of using the BatchNorm layer in two different states, i.e., compare between using running statistics and current statistics. Note that we do this change only in the case of the teacher network. In contrast, the student network always uses current statistics, even at the time of evaluation on the validation set. We do this experiment on CIFAR10 by considering two independent parameters: (i) the state of BatchNorm layers in the teacher network, and (ii) the input that is being fed to both the networks.

We distill a ResNet-34 teacher network pretrained on CIFAR10 into student networks of varying architectures, which are: ResNet-34, ResNet-18, and MobileNetV2. In case of original data, we use standard data augmentations like random crop after padding, random horizontal flip, and normalization. Whereas, while using Gaussian noise, we independently sample pixels for creating a $32 \times 32 \times 3$ image from a standard Gaussian distribution. In both cases, the batch size is 256, and an Adam optimizer with a learning rate of $10^{-3}$ for tuning the parameters of the student network is used[3].

---

[3]Code is available at: `https://github.com/Piyush-555/GaussianDistillation`

Table 1: CIFAR10 distillation in different cases (input fed to the networks and state of BatchNorm in teacher network) across various Student network architectures. The numbers are accuracies obtained on the test data (mean ± standard deviation from three runs). The teacher network here is a ResNet-34, which has an accuracy of 93.29%. The BatchNorm layers in the student model use current statistics during evaluation. Note that RS is running statistics and CS is current statistics.

| Student | ResNet34 | ResNet18 | MobileNetV2 |
|---|---|---|---|
| Supervised | 93.29 | 93.22 | 91.61 |
| Original data + RS (Oracle) | 92.74 ± 0.21 | 92.44 ± 0.05 | 90.57 ± 0.22 |
| Original data + CS | 92.77 ± 0.22 | 92.20 ± 0.1 | 91.44 ± 0.13 |
| Gaussian noise + RS [5, 9, 10, 19] | 13.18 ± 0.21 | 13.49 ± 0.08 | 12.43 ± 0.3 |
| Gaussian noise + CS (Ours) | 87.11 ± 0.23 | 85.98 ± 0.12 | 82.47 ± 0.26 |

Table 2: Results on SVHN, CIFAR100, and Food101 datasets. We see a similar performance of the proposed method as in the previous experiment. Although the gain for CIFAR100 and Food101 is slightly lower, which may be due to the complexity of datasets as the supervised teacher also shows a lower accuracy.

| Dataset | SVHN | CIFAR100 | Food101 |
|---|---|---|---|
| Teacher | ResNet18 | WideResNet-28-10 | ResNet101 |
| Student | MobileNetV2 | WideResNet-16-8 | ResNet18 |
| Teacher supervised | 94.48 | 80.6 | 73.4 |
| Original data + RS (Oracle) | 95.75 | 74.1 | 67.6 |
| Gaussian noise + RS | 45.03 | 1.2 | 0.9 |
| Gaussian noise + CS (Ours) | 92.93 | 65.7 | 54.16 |

The results are shown in Table 1. As expected, the conventional setting of using the original data as input and the teacher with the running statistics computed from the original data is the ideal case, which results in the best performance. Using current statistics does not hinder the learning of the student network either, that is, in the case of original data. However, while using Gaussian noise with running statistics, the activation distribution gets mismatched. On the other hand, using current statistics results in the mitigation of this mismatch. Hence, in the former, the student network is unable to learn at all, whereas in the latter, it achieves an accuracy close to the ideal case, considering that we rely on the random noise[4].

The distribution shifts observed in this experiment are shown in Figure 1b. We use activations of 'avgpool' layer in the teacher network (ResNet-34) to obtain these plots. The figure shows the distribution of average activations across all neurons with respect to different settings discussed above. We observe that while using Gaussian noise, current statistics result in a distribution similar to that of using original data. Also, in the case of Gaussian noise with BatchNorm layers using the running statistics, most neurons have close to zero average activations, implying that they are rarely fired. Their information is, therefore, never used in deciding the outcome, hence that particular information is never learned by the student network. These plots are consistent with the results in Table 1.

## 4.2 Results on Other Datasets

We perform experiments similar to that of CIFAR10 on SVHN, CIFAR100 and Food101 to show the applicability of our method on various datasets and neural network architectures. Since we did not find a considerable variance in the case of CIFAR10, here we report the results on single runs.

The teacher-student pairs are mentioned in Table 2. We use standard data augmentation and a similar setup of hyperparameters, i.e., batches of 256 samples, and optimization by Adam optimizer with a learning rate of $10^{-3}$.

---

[4]An ablation study where we consider some percentage of BatchNorm layers is presented in Appendix A.2.

Table 2 shows the obtained results. We see similar trends as before, except that the Gaussian noise is not at random chance even with running statistics in the case of SVHN. We believe that this is due to relatively lower covariate shift of activation distribution compared to other more sophisticate datasets. For CIFAR100 and Food101 datsets we try out teacher-student architectures which are different from the previous experiments to introduce more variability. Table 2 shows that a large amount of gap between real data-based distillation and data-free distillation is reduced by Gaussian noise and as we see in the next section, the remaining gap can be filled by other mechanisms.

## 4.3 Limited Data

In this experiment, we try to find out the amount of additional data needed for extra efforts after Gaussian noise-based KD. It would be interesting to use the synthetic data for this, however, we rely on samples from the training dataset as proxy due to a large variation in the nature of synthetic data generation approaches. We first pretrain our student model using Gaussian distillation and then finetune it using original data. First, we show how the performance varies with respect to the amount of data used for finetuning, then we briefly show the same for different configurations. We compare our case of pretraining using Gaussian noise against traditional KD using the same fractional amount of data (baseline).

For finetuning, a subset of training data is sampled randomly and a reduced learning rate of $10^{-4}$ is used. Figure 3 (Left) compares the accuracy of the ResNet-18 (student) on CIFAR10 when it is finetuned using the given fraction of training data to its counterpart that is trained from scratch. Further, Table 3 shows the student accuracy for other architectures and datasets with 1% of training data.

We observe a considerable improvement in the student network after finetuning, specifically for MobileNetV2 on CIFAR10 (about 4%). In addition, pretraining using Gaussian noise outperforms the baseline by a large margin.

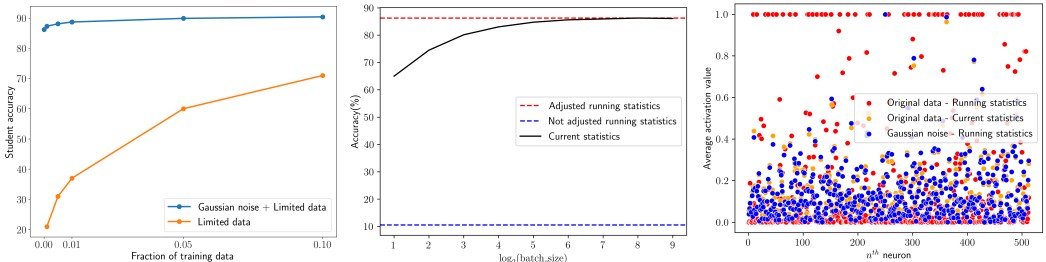

Figure 3: (Left) Varying fractions of data used for finetuning or training from scratch. x-axis denotes the fraction of the training data used and y-axis denotes the corresponding student accuracy. (Middle) Effect of batch size during inference. Note that x-axis represents $\log_2(\text{batch\_size})$. (Right) Scatter plot (clipped to maximum value of 1.0) for average activation for 'avgpool' layer in a student network trained using our approach for CIFAR10.

Table 3: Accuracy of student in various configurations after finetuning using 1% original (training) data. Accuracies are averaged after 3 runs.

| Dataset
Student | CIFAR10
ResNet34 | CIFAR10
MobileNetV2 | SVHN
MobileNetV2 | CIFAR100
WideResNet-16-8 | Food101
ResNet18 |
|---|---|---|---|---|---|
| Limited data | 35.46 | 43.17 | 29.54 | 9.28 | 9.20 |
| Gaussian noise +
Limited data | 89.84 | 86.11 | 94.29 | 69.8 | 61.01 |

### 4.4 BatchNorm State in Student

Here, we explore how the batch size or inference, in general, affects the student network trained using our Gaussian noise-based approach. We use the ResNet-18 student network distilled using our approach from ResNet-34 teacher trained on CIFAR10. We use a sufficient amount of evaluation data to adjust the running statistics of the student's BatchNorm layers, which is 20 batches of size 16 each[5].

After the student is trained using Gaussian noise, its running statistics are not adjusted for test data distribution. Therefore, the accuracy using running statistics is almost at random chance (blue), shown in Figure 3 (Middle). When we use the current statistics and vary the batch size, we see that the accuracy increases with an increase in batch size due to better estimates of $\mu_B$ and $\sigma_B$ (black). Further, we adjust the running statistics using some amount of original data, while keeping the rest of the parameters fixed, considering the case in which the student model needs to infer on a single sample or a batch of non-i.i.d. samples. We observe that the adjusted running statistics yield a performance similar to that of large batch sizes (red).

## 5 Discussion

### 5.1 Limitations

*Absence of BatchNorm:* If there are no BatchNorm layers present in the teacher-student architecture, then the proposed method will not work. In that case, one would need to learn the input data distribution to obtain activation distribution similar to that of original data.

*Training:* As opposed to traditional KD, the teacher in the proposed method is required to use the current statistics while distilling the knowledge to student. In order to correctly calculate these statistics and output correct soft labels, a large enough batch size is required during training period. Results of study regarding batch size are included in Appendix A.4.

*Inference:* As explained in Section 3.3, the student model might be limited by how it can be used during inference. The student network trained using our proposed approach should either be used with a sufficiently large batch of test samples or its running statistics should be adjusted beforehand using a small amount of test data. Without doing so, the student model trained using Gaussian noise does not get adapted to real data distribution, as shown in Figure 3 (Right). The plot shows average activation values for neurons in the 'avgpool' layer of a student network trained using our approach for CIFAR10. The distribution of points for Gaussian noise (blue) and original data with current statistics (orange) is similar. Whereas the points for original data with running statistics (red) either lie close to zero or explode to a very high value (clipped to 1.0).

### 5.2 Rethinking Knowledge Transfer

Recently, [24] explored different types of structured noise (which proves to be far better than Gaussian noise) as a way to learn representations using a specific contrastive loss and questioned the necessity of huge data-driven vision systems. [18] also observed that low-level features are sufficient for performing KD. This paper shows that using just the Gaussian noise (not even low-level features) we can successfully transfer knowledge from one neural network to the other to a large extent. We have not yet explored the possibility of using structured noise, which could prove to be more effective[6]. This challenges the generally accepted notion of requiring synthetic data with representations similar to original data and provides motivation to explore methods similar to ours for tasks such as domain adaptation, transfer learning, etc.

## 6 Conclusions

In this paper, we explore KD solely based on Gaussian noise. We do this by mitigating the covariate shift in the hidden layers of the teacher network, which is caused due to the change in the input data distribution. Further, we show how similar phenomena can occur in the student network, and how

---

[5]Additional results in Appendix A.3.

[6]Preliminary experiments with Dead leaves can be found in Appendix A.5.

we can use a similar approach there as well. The experiments show that student network trained using our approach can achieve a greater performance, in contrast to what prior works have shown for Gaussian noise setting, and more so in the presence of limited data. Future work could be to try Gaussian noise-based distillation with lower input dimensions such that it feed-forwards through both teacher and student network since the Gaussian noise does not contain structural information anyway. Further future work could involve comparing how our approach fares to using unrelated data for KD, trying our approach on structured noise, and extending our findings to other domains such as domain adaptation.

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
