# A  Appendix

## A.1  Toy problem

In order to further understand how running statistics and current statistics with Gaussian noise as input compare against real data, we consider the following toy problem of binary classification of 2D points distributed as concentric circles (real data). We take a simple Multilayer perceptron (MLP) with 2 hidden layers and 2 BatchNorm layers. This is done to restrict the input and embedding space (output from penultimate layer) to 2 dimensions each which can be directly visualized, rather than looking at their distributions. Figure 1 shows the real data on which the MLP is trained, the input Gaussian noise samples, and the embeddings in each of the three cases, respectively from left to right.

We see that, although the distribution of Gaussian noise is considerably different from real data, the embeddings are close to the embeddings of real data while using current statistics as compared to using running statistics.

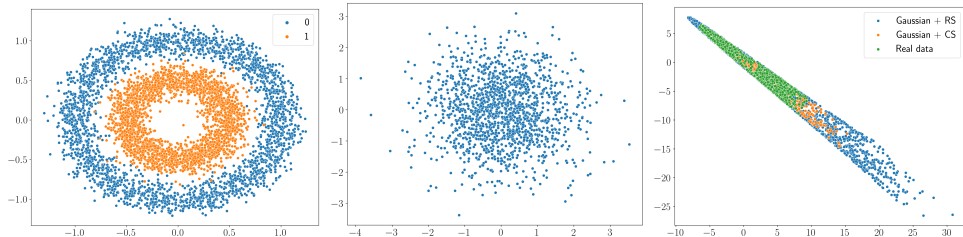

Figure 1: (Left) Circles data on which the MLP is trained. (Middle) Gaussian noise used as input to the trained MLP. (Right) Scatter plot for embeddings in different cases.

## A.2  Handling different number of BatchNorm layers

Here we consider CIFAR10 dataset on which teacher is trained and the teacher-student pair of ResNet34-ResNet18. While performing distillation using Gaussian noise, we randomly choose a given percentage of BatchNorm layers ($\mathcal{P}$) from the teacher network and restrict those layers to use running statistics, whereas the remaining BatchNorm layers use current statistics. We vary the percentage of such layers and report the result in Table 1. We observe that as the percentage of number of BatchNorm layers using running statistics increases, the student accuracy decreases. Note that, the number of BatchNorm layers in the teacher network remains same.

Table 1: Varying percentage of BatchNorm layers ($\mathcal{P}$) that uses running statistics. As the percentage of number of BatchNorm layers using running statistics increases, the student accuracy decreases.

| $\mathcal{P}$ | Student accuracy |
| --- | --- |
| 100 (Running statistics) | 13.49 |
| 90 | 18.26 |
| 75 | 57.73 |
| 50 | 79.54 |
| 25 | 82.24 |
| 0 (Current statistics) | 89.4 |

## A.3  Adjusting the student

From Figure 2, we observe that the student activations for original data with running statistics (which are adapted to Gaussian noise) follows a very different distribution compared to other two, and does not output meaningful information. Neurons in that case either explode or rarely activate. As described in Section 4.4 of the paper, the remedy is to use current statistics while inference or adjust

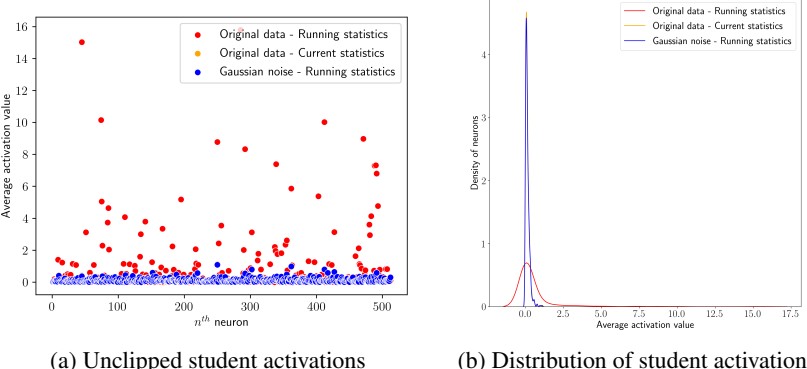

(a) Unclipped student activations  (b) Distribution of student activations

Figure 2: Unclipped scatter plot (linked to Figure 3 (Right) of the paper) and accompanying distribution plot for 'avgpool' layer of the *student network* trained for CIFAR10 using our approach.

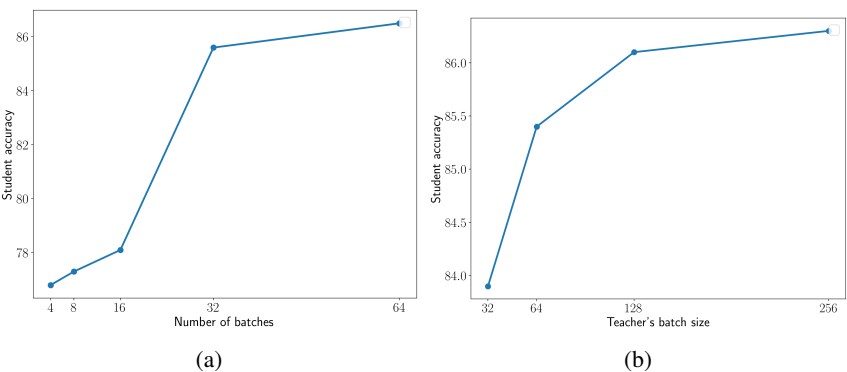

(a)  (b)

Figure 3: (a) Varying number of batches to adjust student's running statistics. We found that the number of batches should be sufficient enough to get good student accuracy. (b) Varying teacher's batch size during distillation. We observe that the larger the batch size, the more accurate the estimates, hence better distillation. The batch size does not need to be too large. Standard batch size for the given dataset should result in good distillation.

the running statistics to make them adapt to real data distribution using a small subset of evaluation data. Figure 3a further shows the number of batches of evaluation data used to adjust the running statistics vs the student's accuracy when those adjusted running statistics are used. Note that the batch size is kept constant at 16. We found that, similar to how the batch size needs to be just large enough while using current statistics, here as well, the number of batches or data in general needs to be sufficient enough to get a good performance out of the student model.

## A.4  Significance of Batch size during distillation

As stated in Section 5.1 of the paper, as opposed to the traditional knowledge distillation, the teacher in our approach needs to rely on the current statistics of the input. In order to get the correct estimates in the BatchNorm layer, the batch size during distillation should be large enough. A comparison is shown in Figure 3b for the case of CIFAR10, where teacher is ResNet-34 and student is ResNet-18. Here, the distillation is performed with the given batch size for both teacher and student models, and the same batch size is used while inference to calculate current statistics.

## A.5  Experiments with Dead Leaves

We conduct additional experiments to understand the potential of the proposed approach. In particular we notice the missing spatial consistency among pixels in the samples randomly drawn from a Gaussian distribution. We, therefore, consider random samples obtained using the Dead leaves

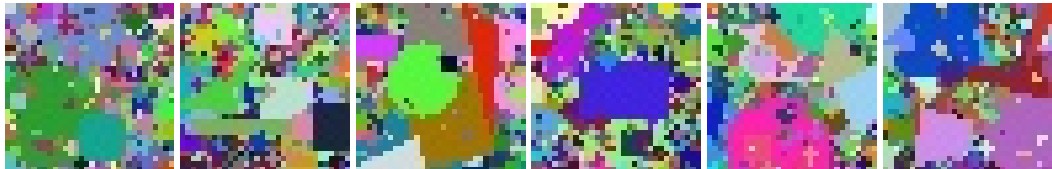

Figure 4: Dead leave samples used for KD during the experiment with CIFAR10

Table 2: CIFAR10 distillation in different cases (input fed to the networks and state of BatchNorm in teacher network) across various Student network architectures. The numbers are accuracy obtained on the test data. The teacher network here is a ResNet-34, which has an accuracy of 93.29%. The BatchNorm layers in the student model use current statistics during evaluation. Note that RS is running statistics and CS is current statistics.

| Student | ResNet34 | ResNet18 | MobileNetV2 | WRN-28-10 | WRN-16-8 |
|---|---|---|---|---|---|
| Original data + RS (Oracle) | 92.74 | 92.44 | 90.57 | 92.41 | 91.32 |
| Gaussian noise + RS | 13.18 | 13.49 | 12.43 | 14.56 | 14.35 |
| Gaussian noise + CS (Ours) | 87.11 | 85.98 | 82.47 | 88.12 | 88.76 |
| Dead leaves + RS | 42.45 | 37.04 | 31.53 | 37.59 | 37.14 |
| Dead leaves + CS (Ours) | 89.7 | 89.4 | 86.94 | 90.75 | 89.96 |

(Shapes)[1] model. Figure 4 shows some of the Dead leave samples. We repeat the distillation experiments on CIFAR10 dataset with Dead leaves samples. We use the same hyperparameter values as the Guassian noise distillation experiments. In addition to ResNet-34, ResNet-18, and MobileNetV2, we consider WRN-28-10 and WRN-16-8 as student networks. Table 2 shows the obtained results. We observe a considerable improvement in the performance of the proposed approach with Dead leave samples. In particular, for MobileNetV2, test accuracy improves more than 4%. Furthermore, for WRN-28-10, the proposed noise-based KD (90.75%) is very close to the real data-based KD (92.41%). Moreover, we consider batch size varying from 32 to 512 and distill ResNet-34, pretrained on CIFAR10, into ResNet-18 using the proposed approach with Dead leaves samples. We observe test accuracy of the student distilled with batch size of 32, 64, 128, 256, and 512 as 88.09%, 88.21%, 89.35%, 89.4%, and 89.19%, respectively. This shows that the further improvements can be obtained by adjusting the nature of random samples.

For further evaluation, we consider DenseNet169-DenseNet121 as teacher-student combination for CIFAR10 dataset, and perform distillation using the proposed approach with Dead leaves samples. Table 3 shows the obtained results where we observe similar improvements as with the previous experiments.

---

[1]https://mbaradad.github.io/learning_with_noise/

Table 3: Results on CIFAR10 teacher - student pair of DenseNet169-DenseNet121.

| Dataset | CIFAR10 |
|---|---|
| Teacher | DenseNet169 |
| Student | DenseNet121 |
| Teacher supervised | 86.23 |
| Original data + RS (Oracle) | 86.23 |
| Gaussian noise + RS | 10.79 |
| Gaussian noise + CS (Ours) | 76.57 |
| Dead leaves + RS | 23.5 |
| Dead leaves + CS (Ours) | 78.33 |

## A.6 Activation distributions

This section contains the distributions of activations of the teacher networks with respect to the model architectures and datasets that they were trained on.

### A.6.1 CIFAR10

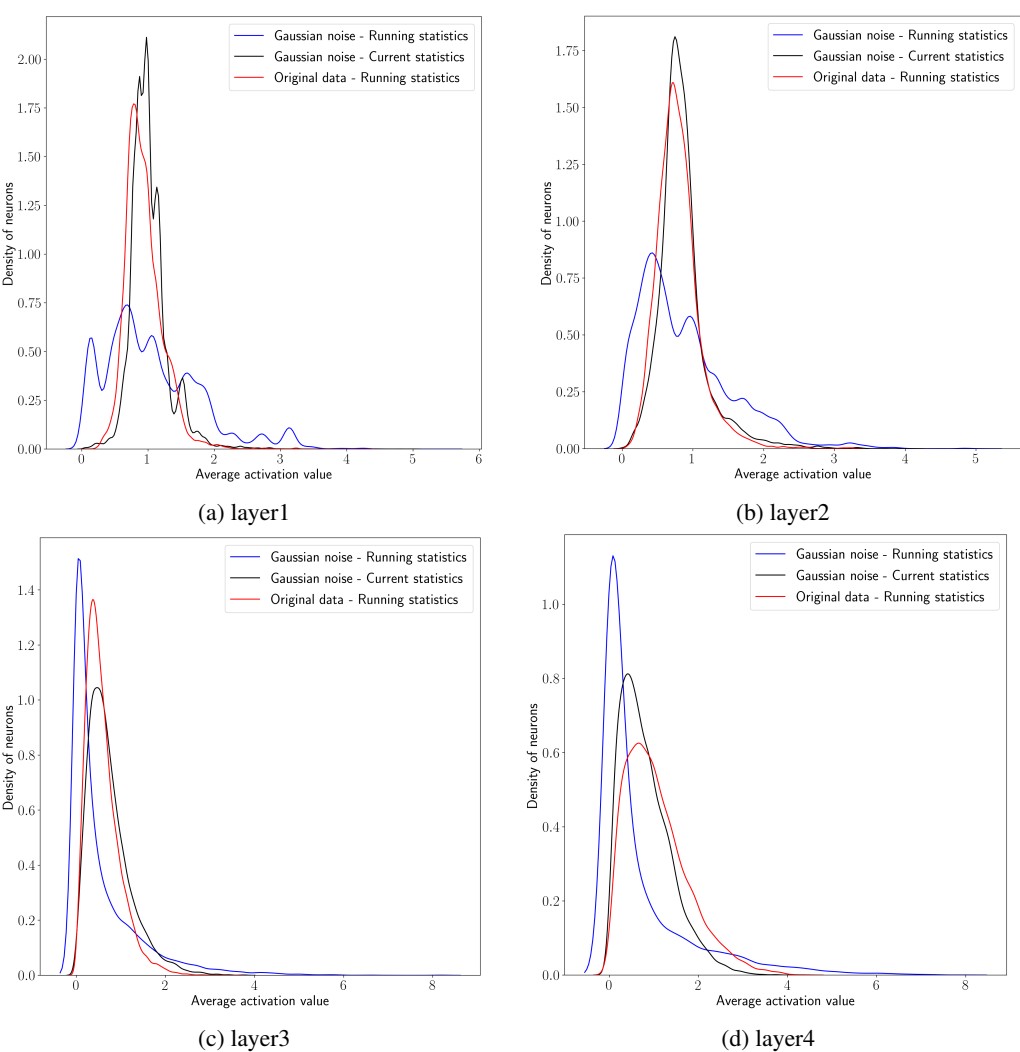

(a) layer1  (b) layer2

(c) layer3  (d) layer4

Figure 5: Distributions for average activation value in ResNet-34 teacher network trained on CIFAR-10 for different layers. Note that the layers follow the standard ResNet naming convention.

In Figure 5, we observe a similar trend in the activation distribution of all layers. The red curve denotes the activation distribution of original data with running statistics, which is the ideal case. However, the trivial use of Gaussian noise with running statistics (blue) results in a significantly different activation distribution. Our proposed approach makes sure that between Gaussian noise with running statistics and Gaussian noise with current statistics (black), the activation distribution of the latter is comparatively similar to the ideal case, thus reducing the shift. We observe a similar trend throughout all the following plots for SVHN (Figure 6), Food101(Figure 7), and CIFAR100 (Figure 8).

 **A.6.2 SVHN**

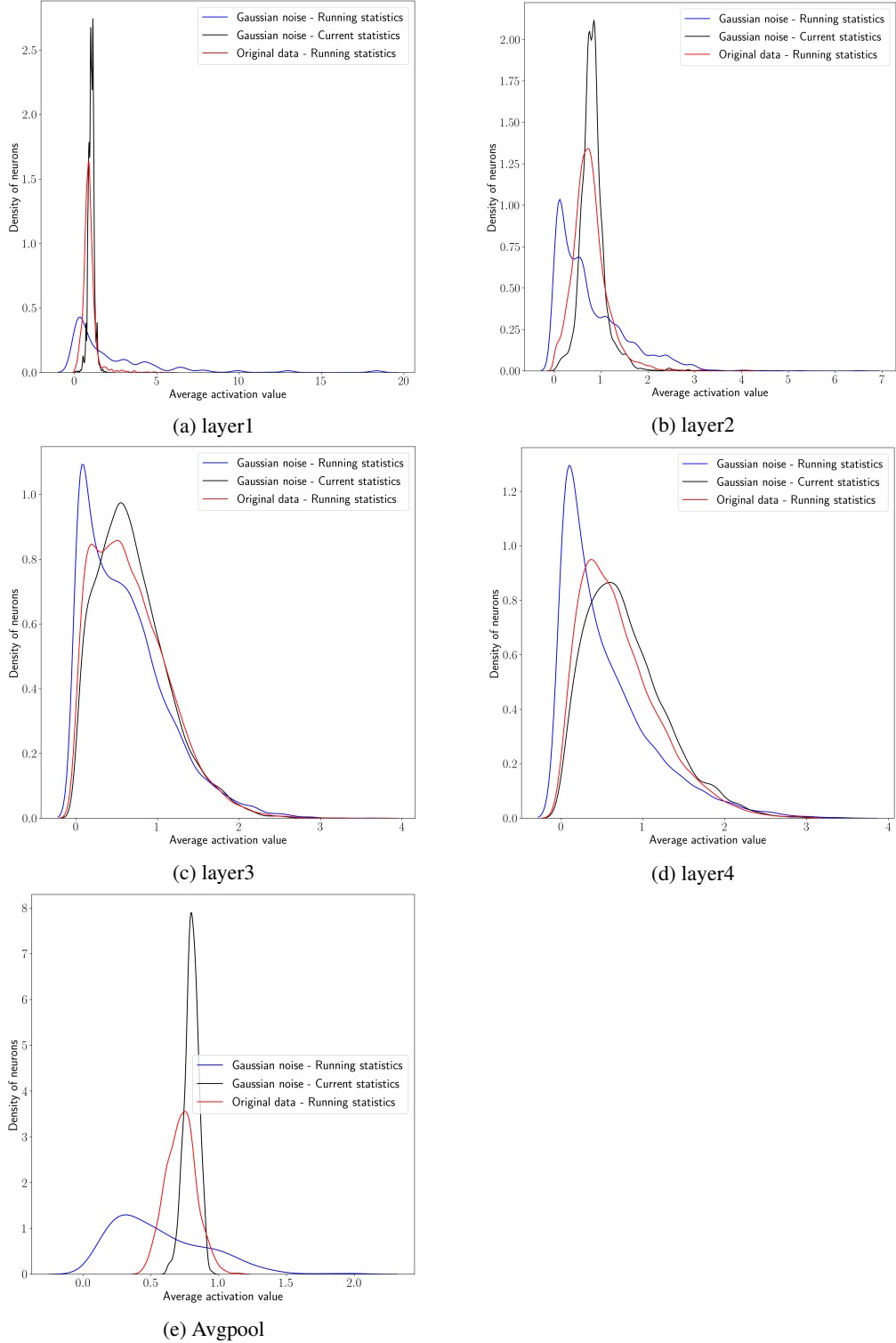

(a) layer1

(b) layer2

(c) layer3

(d) layer4

(e) Avgpool

Figure 6: Distributions for average activation value in ResNet-18 teacher network trained on SVHN for different layers. Note that the layers follow the standard ResNet naming convention.

 **A.6.3   Food101**

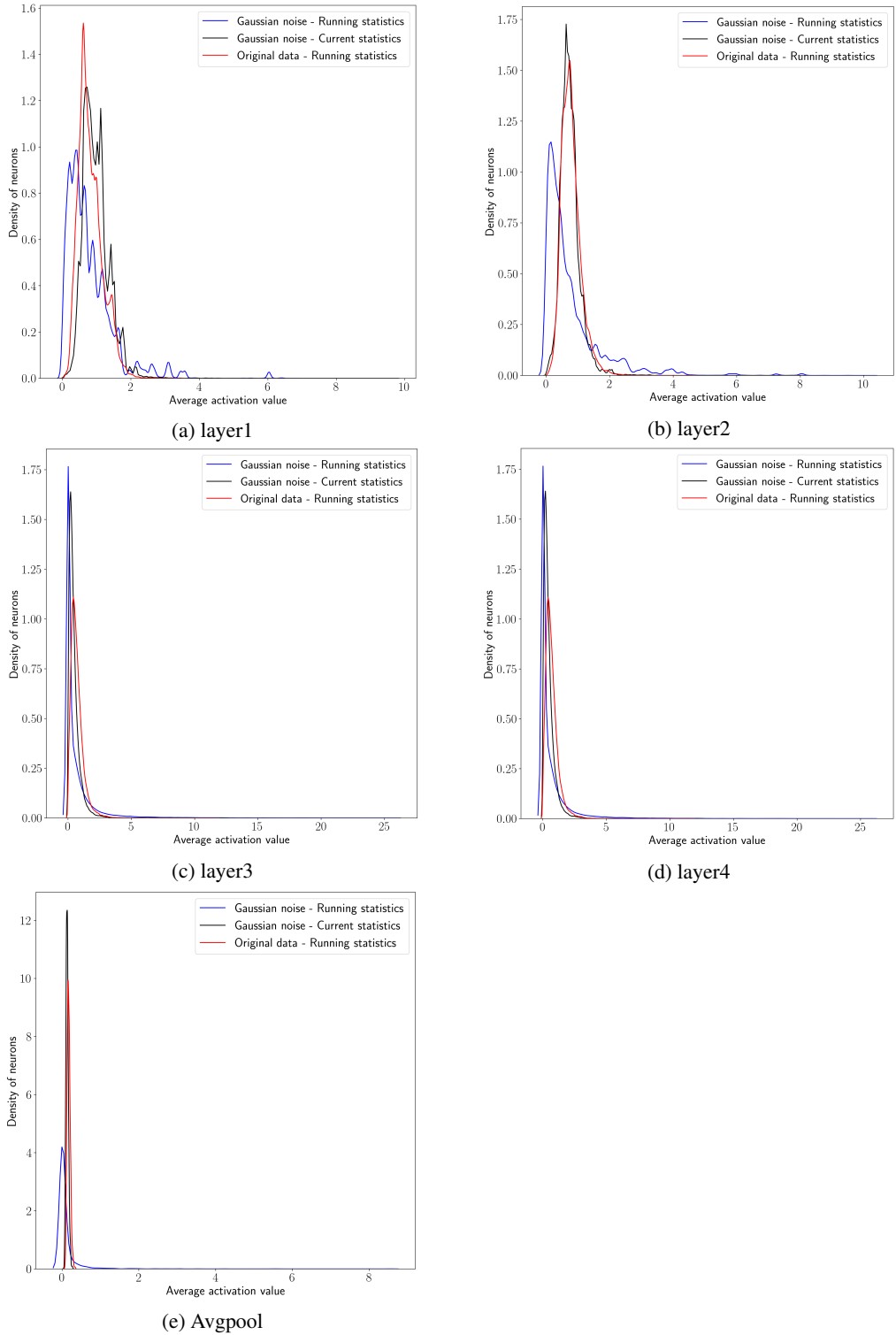

(a) layer1

(b) layer2

(c) layer3

(d) layer4

(e) Avgpool

Figure 7: Distributions for average activation value in ResNet-101 teacher network trained on Food101 for different layers. Note that the layers follow the standard ResNet naming convention.

### A.6.4 CIFAR100

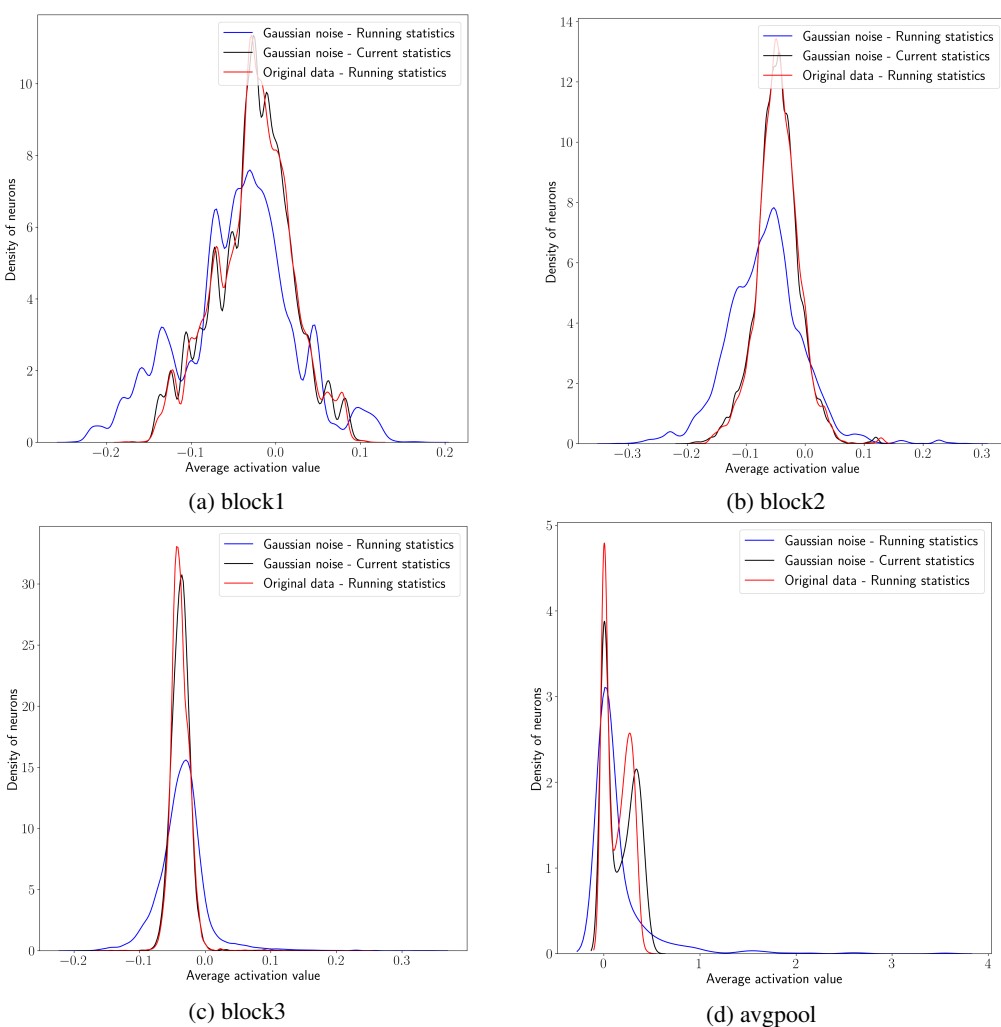

(a) block1      (b) block2

(c) block3      (d) avgpool

Figure 8: Distributions for average activation value in WideResNet-28-10 teacher network trained on CIFAR100 for different layers. Note that the layers follow the standard Wide-ResNet naming convention.

## A.7 Experiment settings

In all of the experiments, we have used standard datasets and model architectures. For Food101, we use PyTorch built-in architectures. For SVHN and CIFAR10, we use a custom implementation[2] of ResNet architectures, which is tailored towards low-resolution images by making the initial kernel shape smaller. For CIFAR100, we use an open-source implementation[3] of Wide-ResNets.

In all cases we have used Adam optimizer with a standard learning rate of 0.001, except for the training of CIFAR100 teacher, where we use SGD with learning rate of 0.1, momentum of 0.9, and weight decay of 0.0005. In all experiments, learning rate scheduler strategy is a standard one to reduce learning rate when a plateau hits. We train teacher and student models in each experiment for 100 epochs and 200 epochs respectively, and wherever not specified we use a batch size of 256.

---

[2]https://github.com/kuangliu/pytorch-cifar
[3]https://github.com/xternalz/WideResNet-pytorch

All of the datasets used have predefined train-test splits with labels available for both, hence the same are used. Note that the train split is used only in case of training the teacher, and test split is used as evaluation data.

**State of BatchNorm.** Handling BatchNorm layers to calculate either running statistics or current statistics is straightforward that can be done by putting the model in one of the two PyTorch modes - *train* and *eval*. Traditionally, during distillation using real data, we put the teacher in *eval* mode where BatchNorm layers use running statistics. For our approach to distill using Gaussian noise, we need the teacher to use current statistics, which is done by putting the teacher in *train* mode. Note that, even though the teacher is in *train* mode, it only allows calculation of current statistics for the BatchNorm layers. No update happens to the teacher's weights as there is no optimizer bound to the teacher's weights.

All of the experiments are performed on a system with 16-core Intel x86 CPU with 128 GB RAM and NVIDIA RTX 3090 GPU.