# OpenReview forum: "Discovering and Overcoming Limitations of Noise-engineered Data-free Knowledge Distillation"
_NeurIPS.cc/2022/Conference — NeurIPS 2022 Accept_

### Official Review · Reviewer_Ao4y · 2022-07-08

**Rating:** 4
**Confidence:** 4
**Soundness:** 3 good
**Presentation:** 2 fair
**Contribution:** 3 good

**Summary:**

The article proposes a method for distillation using Gaussian noise by modifying the distribution of BN layers. The article describes why the accuracy of the previous method is severely degraded and how the distribution can be adjusted to improve the distillation performance.

**Questions:**

See the weaknesses.

**Limitations:**

See the weaknesses.

**Strengths And Weaknesses:**

$\textbf{Strengths}$

1.It proposes a method for distillation using Gaussian noise by modifying the distribution of BN layers；
2. Significantly better than the experimental results of other methods.

$\textbf{Weaknesses}$
1. Fewer other methods of comparison in the experiments;
2. It should try other network structures;
3. Similar to the evaluation phase, if some Gaussian noise is first input to the teacher model before training, is it not necessary to modify the mean and standard deviation of the BN layer?
4. Some experimental details are missing and there are some minor errors in the article that need to be fixed.

---

> ### Author Response · Authors · 2022-08-02
> **For reviewer Ao4y**
>
> We would like to thank the reviewer for constructive feedback. We emphasize that our work identifies the bottleneck in noise-based KD and counters the commonly accepted notion of requiring real or realistic synthesized data for data-free KD. Our proposed approach is a simple yet effective solution to the identified bottleneck, and we believe it would inspire further research in the direction of noise-based data-free KD. Our response to address the raised concerns is as follows.
>
> **1) Fewer other methods of comparison in the experiments**
>
> The primary motive of the paper is to address the limitation of noise-based KD. We, therefore, compared our proposed approach against the prior works that attempted noise-based KD. Further, for a fair comparison between our method and other data-free KD methods, such as synthesized image-based KD, the computational requirements would also be required to take into account along with the student's performance. Moreover, there are many factors like the number of samples synthesised, if the generator is used, and the technique used for image synthesis, which makes the task of measuring the computational requirements non-trivial. Hence, we could not include such approaches for comparison.
>
> **2) It should try other network structures**
>
> We have conducted additional experiments which were feasible in the rebuttal time duration. We consider following additional teacher-student combinations for experiment on the CIFAR10 dataset using Gaussian noise as input.
>         - ResNet34 - WRN-28-10
>         - ResNet34 - WRN-16-8
>         - DenseNet169 - DenseNet121
>
> Moreover, apart from Gaussian noise, we consider random samples obtained using the Dead leave (Shapes) model ["Learning to See by Looking at Noise" by Baradad et al.] for the proposed approach and perform KD for the following teacher-student pairs.
>         - ResNet34 - ResNet34
>         - ResNet34 - ResNet18
>         - ResNet34 - MobileNetV2
>         - ResNet34 - WRN-28-10
>         - ResNet34 - WRN-16-8
>         - DenseNet169 - DenseNet121
>
> We observe a considerable improvement in the performance of the proposed approach with Dead leaves samples. In particular, for MobileNetV2, test accuracy improves by more than 4\%. Similarly, for WRN-28-10, the proposed noise-based KD (90.75\%) is very close to the real data-based KD (92.41\%). This shows that further improvements can be obtained by adjusting the nature of random samples. All the obtained results are included in Appendix A.5 in the supplementary material.
>
> **3) Similar to the evaluation phase, if some Gaussian noise is first input to the teacher model before training, is it not necessary to modify the mean and standard deviation of the BN layer?**
>
> Passing Gaussian noise as input to the teacher model before training will only affect the running statistics of the BatchNorm layers. This would not affect the performance of the proposed approach as it uses the current statistics of the teacher model during distillation.
>
> **4) Some experimental details are missing, and there are some minor errors in the article that need to be fixed.**
>
> We have extended the experimental details in Appendix A.4 in the supplementary material about how we handle BatchNorm layers. We do this by putting the teacher network's BatchNorm layers in one of the PyTorch modes - *eval* for running statistics and *train* for current statistics. Further, we thoroughly checked the paper for minor errors.

---

### Official Review · Reviewer_fgeM · 2022-07-09

**Rating:** 7
**Confidence:** 4
**Soundness:** 3 good
**Presentation:** 3 good
**Contribution:** 3 good

**Summary:**

This paper studies data-free knowledge transfer, specifically a new frontier using only Gaussian Noise as inputs. As known to literature, noise-based data-free KD imposes critical shift in intermediate distributions, and results in very limited efficacy in distillation. The paper proposes a very simple trick to overcome the issue using current batch mean and variances to normalize, as opposed to running stats, and the method seems to work surprisingly well. Method has been explored on CIFAR10, CIFAR100, SVHN, and Food101 datasets over changing network architectures.

**Questions:**

- How will this method scale to deeper networks, say ResNet-50, that has bottleneck blocks instead of basic blocks. Will this cause a difference?
- How will the method scale to ImageNet level analysis?
- How are the related work baselines in Table 1 implemented? Providing more details here can be helpful.

**Limitations:**

In my opinion the paper addresses these fronts well.

**Strengths And Weaknesses:**

Strengths

+ The problem of using Gaussian noise only to data-free distillation is very interesting, as it bypasses the need to synthesize lots of data or train auxiliary networks.
+ The approach is rather simple though, as the main trick is to replace the running stats with batch-wise stats, and has to slightly change the inference flow, that results in small degradation in accuracy. But it's pretty impressive observation that simply doing this trick allows for significant boost in distillation accuracy.
+ I think this paper can inspire deeper analysis of the potential of noise-based KD. The analysis from various fronts are informative.
+ Codes are provided.

Weekness
- It would be very beneficial to observe whether this holds for ImageNet1K-level KD, as the batch-wise stats and running stats discrepancy does impose performance drop, and it would be beneficial to see if the method works for large-scale high-resolution datasets. This can significantly improve the quality of paper.

---

> ### Author Response · Authors · 2022-08-02
> **For reviewer fgeM**
>
> We would like to thank the reviewer for noticing the strengths of our work and indicating that it provides inspiration for a deeper analysis of the potential of noise-based KD. Our response to the concerns is as follows.
>
> **1) It would be very beneficial to observe whether this holds for large-scale high-resolution datasets like ImageNet1K, which can also improve the quality of the paper.**
>
> Unfortunately, due to limited resources, we could not report any numbers on ImageNet. We agree that the discrepancy between batch statistics and running statistics might affect the performance. But, as stated in the paper, this work lays the foundation for noise-based KD, which works very well on CIFAR, as well as Food101, which is a high-resolution dataset. There are many things that can be considered for further improvements, including but not limited to exploring various noises ["Learning to See by Looking at Noise" by Baradad et al.], using our approach at a pretraining phase to reduce the computational complexity of other data-free KD methods, and combining our idea of using current statistics with other data-free KD approaches. To give a quick view into the potential improvements, we added results on data-free KD using Dead leaves images in A.5.
>
> **2) How will this method scale to deeper networks, say ResNet50, that has bottleneck blocks instead of basic blocks?**
>
> We expect a considerable performance gain with deeper networks as well. Our approach relies on the way statistics of BatchNorm layers of the teacher are handled. As shown in section 4.2 of the paper, we considered ResNet101, containing several bottleneck blocks, as the teacher model in the experiment performed on the Food101 dataset. We observed in Table 2 of the paper a large improvement in the noise-based distillation performance when current statistics (54\%) are used as compared to running statistics (0.9\%).
>
> **3) How are the related work baselines in Table 1 implemented?**
>
> All the related works mentioned in Table 1 synthesize pseudo data for KD. These works have reported results on KD using noise, where instead of using synthesized images, noise samples are used. This is done with conventional settings where BatchNorm layers in teacher model are used with running statistics. Implementation is the same as that of a conventional knowledge distillation pipeline, except the input data is sampled from a Gaussian distribution rather than the original (real) dataset. That is, to implement these baselines, we use Gaussian noise as input but keep the teacher in *eval* (PyTorch convention) mode. See more details in A.4.

---

> > ### Comment · Reviewer_fgeM · 2022-08-08
> > **Reviewer comments**
> >
> > The response has addressed my concerns and I think this paper offers insightful inputs to the field. I reside on positive side and keep my score.

---

### Official Review · Reviewer_Ci53 · 2022-07-11

**Rating:** 5
**Confidence:** 5
**Soundness:** 3 good
**Presentation:** 3 good
**Contribution:** 3 good

**Summary:**

This paper proposes a super straightforward approach for data-free knowledge distillation (KD). Instead of generating pseudo samples to simulate the original training images, it uses Gaussian noise samples as the input and proposes to use current statistics rather than running statistics in the BatchNorm layers for knowledge transfer, which greatly improves the performance. The proposed approach is evaluated with several image classification datasets and the results indicates that it outperforms other noise input-based data-free KD approaches.

**Questions:**

Overall, I think this is an interesting paper that proposes a super simple but effective way that uses noise inputs for data-free KD. Compared to traditional KD which puts a lot of effort into sample generation, the proposed approach saves tremendous computation resources. I'd like to see how the authors respond to the cons that I'm concerned about and will consider raising my score accordingly.

**Limitations:**

The authors have adequately discussed the limitations in the manuscript.

**Strengths And Weaknesses:**

Pros:
1. Using noise inputs from a Gaussian distribution is a more efficient way for data-free KD since it saves a lot of computation resources during the pseudo data generation process.
2. The proposed approach is surprisingly easy.
3. Results look good compared to other noise input-based data-free KD approaches.
4. The paper is well written and easy to follow.


Cons:
1. Some of the recent related works of data-free KD are missing:
[1] Nayak, Gaurav Kumar, Konda Reddy Mopuri, Vaisakh Shaj, Venkatesh Babu Radhakrishnan, and Anirban Chakraborty. "Zero-shot knowledge distillation in deep networks." In International Conference on Machine Learning, pp. 4743-4751. PMLR, 2019.
[2] Wang, Zi. "Data-free knowledge distillation with soft targeted transfer set synthesis." In Proceedings of the AAAI Conference on Artificial Intelligence, vol. 35, no. 11, pp. 10245-10253. 2021.

2. Although the proposed approach is surprisingly simple and I agree with the authors that using current statistics is much better than running statistics. However, it is not quite intuitive to imagine why the outputs with current statistics are very close to those from the original training samples because the inputs are quite different. It could be better if the authors could provide more evidence in the ablation study.

3. I see that the performance of the proposed approach performs better than other works that use noise inputs as the baseline, but there is still room for improvement compared to pseudo image synthesized methods [1,2,3].
[3] Yin, Hongxu, Pavlo Molchanov, Jose M. Alvarez, Zhizhong Li, Arun Mallya, Derek Hoiem, Niraj K. Jha, and Jan Kautz. "Dreaming to distill: Data-free knowledge transfer via deepinversion." In Proceedings of the IEEE/CVF Conference on Computer Vision and Pattern Recognition, pp. 8715-8724. 2020.

4. As also discussed in the paper, there are some limitations of the proposed approach: (1) it only works when there are BatchNorm layers in the model, and (2) a large batch needs to be fed into the networks in order to get the statistics. These limitations slightly weaken the contribution of the paper.

---

> ### Author Response · Authors · 2022-08-02
> **For reviewer Ci53**
>
> We would like to thank the reviewer for detailed feedback. Our response to the pointed weaknesses is as follows.\\\\
>
> **1) Missing references.**
>
> We have revised our paper and included the suggested references.
>
> **2) Ablation study to support intuitive understanding of the proposed approach.**
>
> The distributions for average activation value in teacher network for different model architectures and datasets, shown in Appendix A.1 in the supplementary material, demonstrate the matching of activations for different inputs when current statistics are used. This matching at different layers in turn brings the outputs close, even if the inputs are different. In order to further understand how running statistics and current statistics with Gaussian noise as input compare against real data, we consider a toy problem of binary classification of 2D points distributed as concentric circles (real data). We take a simple Multilayer perceptron with two hidden layers, each followed by a BatchNorm layer. We observe that even though the distribution of Gaussian noise is different from real data, their embeddings are considerably closer while using current statistics as compared to using running statistics. More details on this along with the illustrations are included in Appendix A.6.1. This provides additional evidence of matching the outputs for real data and Gaussian noise while using current statistics in BatchNorm layers.
>
> Further, we consider another ablation study in A.6.2, where we use the ResNet34-ResNet18 teacher-student pair for Gaussian noise-based KD. We vary the percentage of BatchNorm layers in the teacher network that uses running statistics. We see a monotonic decrease in the student accuracy as the percentage of BatchNorm layers using running statistics increases. This further proves that using current batch statistics is an important component in noise-engineered KD.
>
> **3) Room for improvement compared to pseudo image synthesized methods.**
>
> We agree that the results presented in the paper show that there is room for improvement compared to pseudo image synthesized methods. We identify one reason for a limited performance is the uncorrelated nature of pixels in the samples randomly drawn from a Gaussian distribution. To understand the potential of the proposed approach, we consider random samples obtained using the Dead leaves (Shapes) model ["Learning to See by Looking at Noise" by Baradad et al.]. We distill ResNet34, pretrained on CIFAR10, into ResNet18 using the proposed approach with Dead leaves samples. We use the same hyperparameter values as the Gaussian noise experiments. The distilled student results in test accuracy of 89.4\%, which is considerably better than the Gaussian noise-based distillation (85.98\%), and also shows the reduction in the performance gap between the proposed and pseudo image synthesized approaches. This suggests that further improvements are possible by adjusting the nature of the random samples. We include more results on distillation using Dead leaves samples for different architectures in Appendix A.5 in the supplementary material. Moreover, we have used standard experiment settings without any hyperparameter tuning. The performance can be further improved with better optimization strategies, such as using sophisticated learning rate schedulers.
>
> **4) Limitations - Needs (1) BatchNorm layers, and (2) Large batch size.**
>
> We agree that there are some limitations of the proposed approach, particularly the requirement of BatchNorm layers. However, the BatchNorm layers are very common in today's deep models and are also a critical requirement to obtain the desired performance in various applications, including data-free KD [for example, "Dreaming to distill: Data-free knowledge transfer via DeepInversion" by Yin et al.]. On the other hand, a large batch size is not a *critical* requirement for the proposed approach. Even the small batch size results in reasonable performance, especially when there exists some spatial consistency in the samples. To understand this better, we consider batch size varying from 32 to 512 and distill ResNet34, pretrained on CIFAR10, into ResNet18 using the proposed approach with Dead leaves samples. We observe test accuracy of the student distilled with batch size of 32, 64, 128, 256, and 512 as 88.09\%, 88.21\%, 89.35\%, 89.4\%, and 89.19\%, respectively.

---

### Author Response · Authors · 2022-08-02
**For all reviewers**

We are thankful to the reviewers for their valuable feedback. We are encouraged that the reviewers find our work interesting, our approach simple and effective, the observations impressive, and the experiments useful. There are some concerns that we try to address individually in the respective responses. In the rebuttal revision, we include two new appendices containing experimental results obtained using Dead leaves samples and ablation studies, respectively. In addition, we corrected minor errors in the paper and incorporated the suggested changes.

---

> ### Author Response · Authors · 2022-08-07
> **For all reviewers**
>
> We thank all the reviewers again for their comments. We hope our responses have been able to address the concerns and looking forward to the feedback on our response. We would be happy to discuss any further issues and respond if there are new comments.

---

### Meta-Review · Area_Chair_FSEg · 2022-08-31

**Recommendation:** Accept
**Confidence:** Certain

**Metareview:**

Reviewers agree that the results presented in this paper are quite solid.  While some reviewers would still like to see a wider range of experiments on different architectures, the consensus seems to be that this paper provides a solid proof of concept for a solution to a very difficult problem.

**Award:**

No

---

### Decision · Program_Chairs · 2022-09-14

Accept